# PROGRESSIVE PURIFICATION FOR INSTANCE-DEPENDENT PARTIAL LABEL LEARNING

## ABSTRACT

*Partial label learning* (PLL) aims to train multi-class classifiers from instances with partial labels (PLs)—a PL for an instance is a set of candidate labels where a *fixed but unknown* candidate is the true label. In the last few years, the *instance-independent* generation process of PLs has been extensively studied, on the basis of which many practical and theoretical advances have been made in PLL, whereas relatively less attention has been paid to the practical setting of *instance-dependent* PLs, namely, the PL depends not only on the true label but the instance itself. In this paper, we propose a theoretically grounded and practically effective approach called *PrOgressive Purification* (POP) for instance-dependent PLL: in each epoch, POP updates the learning model while purifying each PL for the next epoch of the model training by progressively moving out false candidate labels. Theoretically, we prove that POP enlarges the region appropriately fast where the model is *reliable*, and eventually approximates the Bayes optimal classifier with mild assumptions; technically, POP is flexible with arbitrary *losses* and compatible with deep networks, so that the previous advanced PLL losses can be embedded in it and the performance is often significantly improved.

## 1 INTRODUCTION

Over-parameterized deep neural networks owe their popularity much to their ability to (nearly) perfectly memorize large numbers of training examples, and the memorization is known to decrease the generalization error Feldman (2020). On the other hand, scaling the acquisition of examples for training neural networks inevitably introduces non-fully supervised data annotation, a typical example among which is *partial label* Nguyen & Caruana (2008); Cour et al. (2011); Zhang et al. (2016; 2017b); Feng & An (2018); Xu et al. (2019); Yao et al. (2020b); Lv et al. (2020); Feng et al. (2020b); Wen et al. (2021)—a partial label for an instance is a set of candidate labels where a *fixed but unknown* candidate is the true label. *Partial label learning* (PLL) trains multi-class classifiers from instances that are associated with partial labels. It is therefore apparent that some techniques should be applied to prevent memorizing the false candidate labels when PLL resorts to deep learning, and unfortunately, empirical evidence has shown general-purpose regularization cannot achieve that goal Lv et al. (2021).

A large number of deep PLL algorithms have recently emerged that aimed to design regularizers Yao et al. (2020a;b); Lyu et al. (2022) or network architectures Wang et al. (2022a) for PLL data. Further, there are some PLL works that provided theoretical guarantees while making their methods compatible with deep networks Lv et al. (2020); Feng et al. (2020b); Wen et al. (2021); Wu & Sugiyama (2021). We observe that these existing theoretical works have focused on the *instance-independent* setting where the generation process of partial labels is homogeneous across training examples. With an explicit formulation of the generation process, the *asymptotical consistency* Mohri et al. (2018) of the methods, namely, whether the classifier learned from partial labels approximates the Bayes optimal classifier, can be analyzed.

However, the instance-independent process cannot model the real world well since data labeling is prone to different levels of error in tasks of varying difficulty. Intuitively, *instance-dependent* (ID) partial labels should be quite realistic as some poor-quality or ambiguous instances are more difficult to be labeled with an exact true label. Although the instance-independent setting has been extensively studied, on the basis of which many practical and theoretical advances have been made

in PLL, relatively less attention has been paid to the practically relevant setting of ID partial labels. Very recently, one solution has been proposed Xu et al. (2021) which learned directly from ID partial labels, nevertheless, it is still unclear in theory whether the learned classifier is good. Motivated by the above observations, we set out to investigate ID PLL with the aim of proposing a learning approach that is model-independent and theoretically explain when and why the proposed method works.

In this paper, we propose *PrOgressive Purification* (POP), a theoretically grounded PLL framework for ID partial labels. Specifically, we use the observed partial labels to pretrain a randomly initialized classifier (deep network) for several epochs, and then we update both partial labels and the classifier for the remaining epochs. In each epoch, we purify each partial label by moving out the candidate labels for which the current classifier has high confidence of being incorrect, and subsequently we train the classifier with the purified partial labels in the next epoch. As a consequence, the false candidate labels are gradually sifted out and the classification performance of the classifier is improved. We justify POP and outline the main contributions below:

- We propose a novel approach named POP for the ID PLL problem, which purifies the partial labels and refines the classifier iteratively. Extensive experiments validate the effectiveness of POP.
- We prove that POP can be guaranteed to enlarge the region where the model is *reliable* by a promising rate, and eventually approximates the Bayes optimal classifier with mild assumptions. This proof process does *not rely on* the assumption of the instance-independent setting. To the best of our knowledge, this is the first theoretically guaranteed approach for the general ID PLL problem.
- POP is flexible with respect to *losses*, so that the losses designed for the instance-independent PLL problems can be embedded directly. We empirically show that such embedding allows advanced PLL losses can be applied to the ID problem and achieve state-of-the-art learning performance.

## 2   RELATED WORK

In this section, we briefly go through the seminal works in PLL, focusing on the theoretical works and discussing the underlying assumptions behind them.

**Non-deep PLL** There have been substantial non-deep PLL algorithms from the pioneering work Jin & Ghahramani (2003). From a practical standpoint, they have been studied along two different research routes: the identification-based strategy and the average-based strategy. The identification-based strategy purifies each partial label and extracts the true label heuristically in the training phase, so as to identify the true labels Chen et al. (2014); Zhang et al. (2016); Tang & Zhang (2017); Feng & An (2019); Xu et al. (2019). On the contrary, the average-based strategy treats all candidates equally Hüllermeier & Beringer (2006); Cour et al. (2011); Zhang & Yu (2015). On the theoretical side, Liu and Dietterich Liu & Dietterich (2012) analyzed the learnability of PLL by making a *small ambiguity degree condition* assumption, which ensures classification errors on any instance have a probability of being detected. And Cour *et al.* Cour et al. (2011) proposed a consistent approach under the small ambiguity degree condition and a dominance assumption on data distribution (Proposition 5 in Cour et al. (2011)). Liu and Dietterich Liu & Dietterich (2012) proposed a Logistic Stick-Breaking Conditional Multinomial Model to portray the mapping between instances and true labels while assuming the generation of the partial label is independent of the instance itself. It should be noted that the vast majority of non-deep PLL works have only empirically verified the performance of algorithms on small data sets, without formalizing the statistical model for the PLL problem, and therefore even less so for theoretical analysis of when and why the algorithms work.

**Deep PLL** In recent years, deep learning has been applied to PLL and has greatly advanced the practical application of PLL. Yao *et al.* Yao et al. (2020a;b) and Lv *et al.* Lv et al. (2020) proposed learning objectives that are compatible with stochastic optimization and thus can be implemented by deep networks. Soon Feng *et al.* Feng et al. (2020b) formalized the first generation process for PLL. They assumed that given the latent true label, the probability of all incorrect labels being added into the candidate label set is uniform and independent of the instance. Thanks to the uniform generation process, they proposed two provably consistent algorithms. Wen *et al.* Wen et al. (2021) extended the uniform one to the *class-dependent* case, but still keep the instance-independent as-

sumption unchanged. In addition, a new paradigm called complementary label learning Ishida et al. (2017); Yu et al. (2018); Ishida et al. (2019); Feng et al. (2020a) has been proposed that learns from instances equipped with a complementary label. A complementary label specifies the classes to which the instance does not belong, so it can be considered to be an inverted PLL problem. However, all of them made the instance-independent assumption for analyzing the statistic consistency. Wu and Sugiyama Wu & Sugiyama (2021) proposed a framework that unifies the formalization of multiple generation processes under the instance-independent assumption. Wang *et al.* Wang et al. (2022a) proposed a data-augmentation-based framework to disambiguate partial labels with contrastive learning. Zhang *et al.* Zhang et al. (2021a) exploited the class activation value to identify the true label in candidate label sets.

Very recently, some researchers are beginning to notice a more general setting—ID PLL. Learning with the ID partial labels is challenging, and all instance-independent approaches cannot handle the ID PLL problem directly. Specifically, the theoretical approaches mentioned above utilize mainly the *loss correction* technique, which corrects the prediction or the loss of the classifier using a *prior or estimated knowledge* of data generation processes, i.e., a set of parameters controlling the probability of generating incorrect candidate labels, or it is often called transition matrix Patrini et al. (2017). The transition matrix can be characterized fixedly in the instance-independent setting since it does not need to include instance-level information, a condition that does not hold in ID PLL. Furthermore, it is *ill-posed* to estimate the transition matrix by only exploiting partially labeled data, i.e., the transition matrix is unidentifiable Xia et al. (2020). Therefore, some new methods should be proposed to tackle this issue. Xu *et al.* Xu et al. (2021) introduced a solution that infers the latent label posterior via variational inference methods Blei et al. (2017), nevertheless, its effectiveness would be hardly guaranteed. In this paper, we propose POP for the ID PLL problem and theoretically prove that the learned classifier approximates well to the Bayes optimal.

## 3 PROPOSED METHOD

### 3.1 PRELIMINARIES

First of all, we briefly introduce some necessary notations. Consider a multi-class classification problem of $c$ classes. Let $\mathcal{X} = \mathbb{R}^q$ be the $q$-dimensional instance space and $\mathcal{Y} = \{1, 2, \ldots, c\}$ be the label space with $c$ class labels. In supervised learning, let $p(\boldsymbol{x}, y)$ be the underlying "clean" distribution generating $(\boldsymbol{x}, y^{\boldsymbol{x}}) \in \mathcal{X} \times \mathcal{Y}$ from which $n$ i.i.d. samples $\{(\boldsymbol{x}_i, y^{\boldsymbol{x}_i})\}_{i=1}^n$ are drawn.

In PLL, there is a *partial label space* $\mathcal{S} := \{S | S \subseteq \mathcal{Y}, S \neq \emptyset\}$ and the PLL training set $\mathcal{D} = \{(\boldsymbol{x}_i, S_i) | 1 \leq i \leq n\}$ is sampled independently and identically from a "corrupted" density $\tilde{p}(\boldsymbol{x}, S)$ over $\mathcal{X} \times \mathcal{S}$. It is generally assumed that $p(\boldsymbol{x}, y)$ and $p(\boldsymbol{x}, S)$ have the same marginal distribution of instances $p(\boldsymbol{x})$. Then the *generation process* of partial labels can thus be formalized as $p(S|\boldsymbol{x}) = \sum_y p(S|\boldsymbol{x}, y)p(y|\boldsymbol{x})$. We define the probability that, given the instance $\boldsymbol{x}$ and its class label $y^{\boldsymbol{x}}$, $j$-label being included in its partial label as the *flipping probability*:

$$\xi^j(\boldsymbol{x}) = p(j \in S|\boldsymbol{x}, y^{\boldsymbol{x}}), \ \forall j \in \mathcal{Y},$$

The key definition in PLL is that the latent true label of an instance is always one of its candidate label, i.e., $\xi^{y^{\boldsymbol{x}}}(\boldsymbol{x}) = 1$.

We consider use deep models by the aid of an inverse link function Reidand & Williamson (2010) $\phi : \mathbb{R}^c \rightarrow \Delta^{c-1}$ where $\Delta^{c-1}$ denotes the $c$-dimensional simplex, for example, the softmax, as learning model in this paper. Then the goal of supervised multi-class classification and PLL is the same: a scoring function $f : \mathcal{X} \mapsto \Delta^{c-1}$ that can make correct predictions on unseen inputs. Typically, the classifier takes the form:

$$h(\boldsymbol{x}) = \arg \max_{j \in \mathcal{Y}} f_j(\boldsymbol{x}).$$

The Bayes optimal classifier $h^\star$ (learned using supervised data) is the one that minimizes the risk w.r.t the 0-1 loss (or some classification-calibrated loss Bartlett et al. (2006)), i.e.,

$$h^\star = \arg \min_h \mathcal{R}_{01} = \arg \min_h \mathbb{E}_{(\boldsymbol{X}, Y) \sim p(\boldsymbol{x}, y)} \left[ \mathbf{1}_{\{h(\boldsymbol{X}) \neq Y\}} \right].$$

For *strictly proper losses* Gneiting & Raftery (2007), the scoring function $f^*$ recovers the class-posterior probabilities, i.e., $f^\star(\boldsymbol{x}) = p(y|\boldsymbol{x}), \forall \boldsymbol{x} \in \mathcal{X}$. When the supervision information available

is partial label, the PLL risk under $\tilde{p}(\boldsymbol{x}, S)$ w.r.t. a suitable *PLL loss* $\mathcal{L} : \mathbb{R}^k \times \mathcal{S} \rightarrow \mathbb{R}^+$ is defined as

$$\tilde{\mathcal{R}} = \mathbb{E}_{(\boldsymbol{X}, S) \sim \tilde{p}(\boldsymbol{x}, S)} \big[ \mathcal{L}(h(\boldsymbol{X}), S) \big].$$

Minimizing $\tilde{\mathcal{R}}$ induces the classifier and it is desirable that the minimizer approach $h^\star$. In addition, let $o = \arg\max_{j \neq y^{\boldsymbol{x}}} p(y = j | \boldsymbol{x})$ be the class label with the second highest posterior possibility among all labels.

## 3.2 OVERVIEW

In the latter part of this section, we will introduce a concept *pure level set* as the region where the model is *reliable*. We prove that given a tiny reliable region, one could progressively enlarge this region and improves the model with a sufficient rate by disambiguating the partial labels. Motivated by the theoretical results, we propose an approach POP that works by progressively purifying the partial labels to move out the false candidate labels, and eventually the learned classifier could approximate the Bayes optimal classifier.

POP employs the observed partial labels to pre-train a randomly initialized classifier for several epochs, and then updates both partial labels and the classifier for the remaining epochs. We start with a warm-up period, in which we train the predictive model with a well-defined PLL loss Lv et al. (2020). This allows us to attain a reasonable predictive model before it starts fitting incorrect labels Zhang et al. (2017a). After the warm-up period, we iteratively purify each partial label by moving out the candidate labels for which the current classifier has high confidence of being incorrect, and subsequently we train the classifier with the purified partial labels in the next epoch. After the model has been fully trained, the predictive model can perform prediction for unseen instances.

## 3.3 THE POP METHOD

We assume that the hypothesis class $\mathcal{H}$ is sufficiently complex (and deep networks could meet this condition), such that the approximation error equals zero, i.e., $\arg\min_h \mathcal{R} = \arg\min_{h \in \mathcal{H}} \mathcal{R}$ and we have enough training data i.e., $n \rightarrow \infty$. The classifier is able to at least approximate the Bayes optimal classifier $h^\star$ and the gap between the learned $f(\boldsymbol{x})$ and the the scoring function $f^\star(\boldsymbol{x})$ corresponding to $h^\star$ is determined by the inconsistency between incorrect candidate labels and output of the Bayes optimal classifier.

For two instance $\boldsymbol{x}$ and $\boldsymbol{z}$ that satisfy $p(y^{\boldsymbol{z}} | \boldsymbol{z}) - p(o | \boldsymbol{z}) \geq p(y^{\boldsymbol{x}} | \boldsymbol{x}) - p(o | \boldsymbol{x})$, i.e., the margin between the posterior of ground-truth label $p(y^{\boldsymbol{z}} | \boldsymbol{z})$ and the second highest posterior possibility $p(o | \boldsymbol{z})$ is larger than that in point $\boldsymbol{x}$, the indicator function $\left[ \mathbf{1}_{\{j \neq h^\star(\boldsymbol{z})\}} \middle| p(y^{\boldsymbol{z}} | \boldsymbol{z}) - p(o | \boldsymbol{z}) \geq p(y^{\boldsymbol{x}} | \boldsymbol{x}) - p(o | \boldsymbol{x}), j \in S_{\boldsymbol{z}} \right]$ equals 1 if the candidate label $j$ of $\boldsymbol{z}$ is inconsistent with the output of the optimal Bayes classifier $h^\star(\boldsymbol{z})$. Then, the gap between $f_j(\boldsymbol{x})$ and $f_j^\star(\boldsymbol{x})$, i.e., the approximation error of the classifier, could be controlled by the inconsistency between the incorrect candidate labels and the output of the Bayes optimal classifier $h^\star$ for all the instances $\boldsymbol{z}$. Therefore, we assume that there exist constants $\alpha, \epsilon < 1$, such that for $f(\boldsymbol{x})$,

$$|f_j(\boldsymbol{x}) - f_j^\star(\boldsymbol{x})| \leq \alpha \mathbb{E}_{(\boldsymbol{z}, S) \sim \tilde{p}(\boldsymbol{z}, S)} \left[ \mathbf{1}_{\{j \neq h^\star(\boldsymbol{z})\}} \middle| p(y^{\boldsymbol{z}} | \boldsymbol{z}) - p(o | \boldsymbol{z}) \geq p(y^{\boldsymbol{x}} | \boldsymbol{x}) - p(o | \boldsymbol{x}), j \in S_{\boldsymbol{z}} \right] + \frac{\epsilon}{6}$$
(1)

where the scoring function $f^*$ corresponding to $h^*$ on *strictly proper losses* Gneiting & Raftery (2007) recovers the class-posterior probabilities, i.e., $f_j^\star(\boldsymbol{x}) = p(y = j | \boldsymbol{x})$. In addition, for the probability density function $d(u)$ of cumulative distribution function $D(u) = P_{\boldsymbol{x} \sim p(\boldsymbol{x}, y)}(u(\boldsymbol{x}) \leq u)$ where $0 \leq u \leq 1$ and the margin $u(\boldsymbol{x}) = p(y^{\boldsymbol{x}} | \boldsymbol{x}) - p(o | \boldsymbol{x})$. we assume that there exist constants $c_\star, c^\star > 0$ such that $c_\star < d(u) < c^\star$. Then, the worst-case density-imbalance ratio is denoted by $l = \frac{c^\star}{c_\star}$. As the flipping probability of the incorrect label in the instance-dependent generation process is related to its posterior probability, we assume that there exists a constant $t > 0$ such that:

$$\xi^j(\boldsymbol{x}) \leq p(y = j | \boldsymbol{x}) t. \tag{2}$$

Motivated by the pure level set in binary classification Zhang et al. (2021b), we define the pure level set in instance-dependent PLL, i.e., the region where the model is reliable:

**Definition 1** *(Pure $(e, f)$-level set). A set $L(e) := \{x \| p(y^x|x) - p(o|x) | \geq e\}$ is pure for $f$ if $y^x = \arg\max_j f_j(x)$ for all $x \in L(e)$.*

Assume that there exists a set $L(e)$ for all $x \in L(e)$ which satisfies $y^x = \arg\max_j f_j(x)$, we have

$$\mathbb{E}_{(z,S) \sim \tilde{p}(z,S)} \left[ \mathbf{1}_{\{j \neq h^\star(z)\}} \Big| p(y^z|z) - p(o|z) \geq p(y^x|x) - p(o|x), \ j \in S_z \right] = 0 \quad (3)$$

which means that there is a tiny region $L(e) := \{x \| p(y^x|x) - p(o|x) | \geq e\}$ where the model $f$ is reliable.

Let $e_{\text{new}}$ be the new boundary and $\frac{\epsilon}{6l\alpha}(p(y^x|x) - e) \leq e - e_{\text{new}} \leq \frac{\epsilon}{3l\alpha}(p(y^x|x) - e)$. As the probability density function $d(u)$ of the margin $u(x) = p(y^x|x) - p(o|x)$ is bounded by $c_\star < d(u) < c^\star$, we have the following result for $x$ that satisfies $e > p(y^x|x) - p(o|x) \geq e_{\text{new}}$ [1]:

$$\mathbb{E}_{(z,S) \sim \tilde{p}(z,S)} \left[ \mathbf{1}_{\{j \neq h^\star(z)\}} \Big| p(y^z|z) - p(o|z) \geq p(y^x|x) - p(o|x), \ j \in S_z \right] \leq \frac{\epsilon}{3\alpha}. \quad (4)$$

Combining Eq. (1) and Eq. (4), there is

$$|f_j(x) - f_j^\star(x)| \leq \frac{\epsilon}{2}. \quad (5)$$

Denote by $m = \arg\max_j f_j(x)$ the label with the highest posterior probability for the current prediction. If $f_m(x) - f_{j \neq m}(x) \geq e + \epsilon$, we have [2]

$$p(y^x|x) \geq p(y = j|x) + e \quad (6)$$

which means that the label $j$ is incorrect label. Therefore, we could move the label $j$ out from the candidate label set to disambiguate the partial label, and then refine the learning model with the partial label with less ambiguity. In this way, we would move one step forward by trusting the model with the tiny reliable region with following theorem.

We start with a warm-up period, as the classifier is able to attain reasonable outputs before fitting label noise Zhang et al. (2017a). Note that the warm-up training is employed to find a tiny reliable region and the ablation experiments show that the performance of POP does not rely on the warm-up strategy. The predictive model $\theta$ could be trained on partially labeled examples by minimizing any PLL loss function. Here we adopt PRODEN loss Lv et al. (2020) to to find a tiny reliable region:

$$\mathcal{L}_{PLL} = \sum_{i=1}^{n} \sum_{j=1}^{c} w_{ij} \ell(f_j(x_i), S_i). \quad (7)$$

Here, $\ell$ is the cross-entropy loss and the weight $w_{ij}$ is initialized with with uniform weights and then could be tackled simply using the current predictions for slightly putting more weights on more possible labels Lv et al. (2020):

$$w_{ij} = \begin{cases} f_j(x_i) / \sum_{j \in S_i} f_j(x_i) & \text{if } j \in S_i \\ 0 & \text{otherwise} \end{cases} \quad (8)$$

**Theorem 1** *Assume that we have enough training data($n \to \infty$) and there is a pure $(e, f)$-level set where $x \in L(e)$ can be correctly classified by $f$. For each $x$ and $\forall j \in S$ and $j \neq m$, if $f_m(x) - f_j(x) \geq e + \epsilon$, we move out label $j$ from the candidate label set and then update the candidate label set as $S_{new}$. Then the new classifier $f_{new}(x)$ is trained on the updated data with the new distribution $\tilde{p}(x, S_{new})$. Let $e_{new}$ be the minimum boundary that $L(e_{new})$ is pure for $f_{new}$. Then, we have*

$$p(y^x|x) - e_{new} \geq (1 + \frac{\epsilon}{6\alpha l})(p(y^x|x) - e).$$

The detailed proof can be found in Appendix A.1. Theorem 1 shows that the purified region $\gamma = p(y^x|x) - e$ would be enlarged by at least a constant factor with the given purification strategy.

---

[1]More details could be found in Appendix A.1.

[2]More details could be found in Appendix A.2.

---

**Algorithm 1** POP Algorithm

---

**Input**: The PLL training set $\mathcal{D} = \{(\boldsymbol{x}_1, S_1), ..., (\boldsymbol{x}_n, S_n)\}$, initial threshold $e_0$, end threshold $e_{\text{end}}$, total round $R$, step-size $e_s$;

 1: Initialize the predictive model $\boldsymbol{\theta}$ by warm-up training with the PLL loss Eq. 7, and threshold $e = e_0$;
 2: **for** $r = 1, ..., R$ **do**
 3:     Train the predictive model $f$ on $\mathcal{D}$;
 4:     **for** $i = 1, ..., n$ **do**
 5:         **for** $j \in S_i$ **do**
 6:             **if** $f_{m_i}(\boldsymbol{x}_i) - f_j(\boldsymbol{x}_i) \geq e + \epsilon$ **then**
 7:                 Purify the incorrect label $j$ by removing it from the candidate label set $S_i$;
 8:             **end if**
 9:         **end for**
10:     **end for**
11:     **if** $e \leq e_{\text{end}}$, and there is no purification for any candidate label set **then**
12:         Decrease $e$ with step-size $e_s$;
13:     **end if**
14: **end for**

**Output**: The final predictive model $f$

---

After the warm-up period, the classifier could be employed for purification. According to Theorem 1, we could progressively move out the incorrect candidate label with the continuously strict bound, and subsequently train an effective classifier with the purified labels with the PLL loss Lv et al. (2020) since the PLL loss Lv et al. (2020) is model-independent and could operates in a mini-batched training manner to update the model with the labeling-confidence weight. Specifically, we set a high threshold $e_0$ and calculate the difference $f_m(\boldsymbol{x}_i) - f_j(\boldsymbol{x}_i)$ for each candidate label. If there is a label $j$ for $\boldsymbol{x}_i$ satisfies $f_m(\boldsymbol{x}_i) - f_j(\boldsymbol{x}_i) \geq e_0$, we move out it from the candidate label set and update the candidate label set. We depart from the theory by reusing the same fixed dataset over and over, but the empirics are reasonable.

If there is no purification for all partial labels, we begin to decrease the threshold $e$ and continue the purification for improving the training of the model. In this way, the incorrect candidate labels are progressively removed from the partial label round by round, and the performance of the classifier is continuously improved. The algorithmic description of POP is shown in Algorithm 1.

Then we prove that if there exists a pure level set for an initialized model, our proposed approach can purify incorrect labels and the classifier $f$ will finally match the Bayes optimal classifier $h$ after sufficient rounds $R$ under the instance-dependence PLL setting .

**Theorem 2** *For any flipping probability of each incorrect label $\xi^j(\boldsymbol{x})$, define $e_0 = \frac{(1+t)\alpha + \frac{\epsilon}{6}}{1+\alpha}$. And for a given function $f_0$ there exists a level set $L(e_0)$ which is pure for $f_0$. If one runs purification in Theorem 1 with enough traing data ($n \rightarrow \infty$) starting with $f_0$ and the initialization: (1) $e_0 \geq \frac{(1+t)\alpha + \frac{\epsilon}{6}}{1+\alpha}$, (2) $R \geq \frac{6l}{\epsilon} \log(\frac{1-\epsilon}{\frac{1}{c} - e_0})$, (3)$e_{end} \geq \epsilon$, then we have:*

$$\mathbb{P}_{\boldsymbol{x} \sim D}[y_{f_{final}(\boldsymbol{x})} = h^\star(\boldsymbol{x})] \geq 1 - c^\star \epsilon$$

The proof of Theorem 2 is provided in Appendix A.3. According to Theorem 2, the learned classifier under the instance-dependent PLL setting will be consistent with the Bayes optimal classifier eventually. Theorem 2 shows that the classifier can be guaranteed to eventually approximate the Bayes optimal classifier.

## 4 EXPERIMENTS

### 4.1 DATASETS

We adopt five widely used benchmark datasets including MNIST LeCun et al. (1998), Kuzushiji-MNIST Clanuwat et al. (2018), Fashion-MNIST Xiao et al. (2017), CIFAR-10 Krizhevsky & Hinton

Table 1: Classification accuracy (mean±std) of each comparing approach on benchmark datasets corrupted by the ID generation process.

|  | MNIST | Kuzushiji-MNIST | Fashion-MNIST | CIFAR-10 | CIFAR-100 |
|---|---|---|---|---|---|
| POP | **99.28±0.02%** | **91.09±0.14%** | **96.93±0.07%** | **93.00±0.26%** | **71.82±0.08%** |
| VALEN | 99.03±0.02% | 90.15±0.02% | 96.31±0.12% | 92.01±0.09% | 71.48±0.12% |
| RCR | 98.81±0.07% | 90.62±0.22% | 96.64±0.10% | 86.11±0.43% | 71.07±0.25% |
| PICO | 98.76±0.04% | 88.87±0.06% | 94.83±0.17% | 89.35±0.17% | 66.30±0.24% |
| PRODEN | 99.01±0.02% | 90.48±0.14% | 96.14±0.07% | 78.87±0.26% | 55.59±0.08% |
| RC | 99.09±0.09% | 90.56±0.14% | 96.17±0.08% | 80.13±0.14% | 56.41±0.17% |
| CC | 99.08±0.10% | 90.40±0.20% | 96.12±0.10% | 76.17±0.11% | 56.48±0.06% |
| LW | 98.98±0.05% | 89.82±0.2% | 93.23±0.08% | 43.16±0.63% | 49.63±0.12% |
| CAVL | 98.95±0.05% | 87.85±0.06% | 95.84±0.06% | 75.41±4.77% | 58.17±0.11% |
| CLPL | 98.83±0.05% | 90.21±0.08% | 93.18±0.08% | 51.61±0.39% | 30.84±0.40% |

Table 2: Classification accuracy (mean±std) of each comparing approach on the real-world datasets.

|  | Lost | BirdSong | MSRCv2 | Mirflickr | Malagasy | Soccer Player | Yahoo!News |
|---|---|---|---|---|---|---|---|
| POP | **78.57±0.45%** | **74.47±0.36%** | 45.86±0.28% | **61.09±0.10%** | **72.29±0.33%** | 54.48±0.10% | **66.38±0.07%** |
| VALEN | 76.87±0.86% | 73.39±0.26% | **49.97±0.43%** | 59 13±0.12% | 69.44±0.06% | 55.81±0.10% | 66.26±0.13% |
| PRODEN | 76.47±0.25% | 73.44±0.12% | 45.10±0.16% | 59.59±0.52% | 69.34±0.09% | 54.05±0.15% | 66.14±0.10% |
| RC | 76.26±0.46% | 69.33±0.32% | 49.47±0.43% | 58.93±0.10% | 70.69±0.14% | **56.02±0.59%** | 63.51±0.20% |
| CC | 63.54±0.25% | 69.90±0.58% | 41.50±0.44% | 58.81±0.54% | 69.53±0.34% | 49.07±0.36% | 54.86±0.48% |
| LW | 73.13±0.32% | 51.45±0.26% | 49.85±0.49% | 54.50±0.81% | 59.34±0.25% | 50.24±0.45% | 48.21±0.29% |
| CAVL | 73.96±0.51% | 69.63±0.93% | 46.62±1.29% | 57.13±0.10% | 65.82±0.06% | 52.92±0.40% | 60.97±0.13% |
| CLPL | 63.39±0.12% | 62.90±3.33% | 37.8±0.71% | 58.87±0.10% | 64.25±0.29% | 48.23±0.03% | 49.42±0.13% |

(2009), CIFAR-100 Krizhevsky & Hinton (2009). These datasets are manually corrupted into ID partially labeled versions. Specifically, we set the flipping probability of each incorrect label corresponding to an instance $x$ by using the confidence prediction of a neural network trained using supervised data parameterized by $\hat{\theta}$ Xu et al. (2021). The flipping probability $\xi^j(x) = \frac{f_j(x;\hat{\theta})}{\max_{j \in \bar{Y}} f_j(x;\hat{\theta})}$, where $\bar{Y}_i$ is the set of all incorrect labels except for the true label of $x_i$. The average number of candidate labels (avg. #CLs) for each benchmark dataset corrupted by the ID generation process is recorded in Appendix A.4.

In addition, five real-world PLL datasets which are collected from different application domains are used, including Lost Cour et al. (2011), Soccer Player Zeng et al. (2013), Yahoo!News Guillaumin et al. (2010), MSRCv2 Liu & Dietterich (2012), and BirdSong Briggs et al. (2012). The average number of candidate labels (avg. #CLs) for each real-world PLL dataset is also recorded in Appendix A.4.

## 4.2 BASELINES

The performance of POP is compared against five deep PLL approaches:

- PRODEN Lv et al. (2020): A progressive identification approach which approximately minimizes a risk estimator and identifies the true labels in a seamless manner;
- RC Feng et al. (2020b): A risk-consistent approach which employs the loss correction strategy to establish the true risk by only using the partially labeled data;
- CC Feng et al. (2020b): A classifier-consistent approach which also uses the loss correction strategy to learn the classifier that approaches the optimal one;
- VALEN Yao et al. (2020a): An ID PLL approach which recovers the latent label distribution via variational inference methods;
- LW Wen et al. (2021): A risk-consistent approach which proposes a leveraged weighted loss to trade off the losses on candidate labels and non-candidate ones.
- CAVL Zhang et al. (2021a): A progressive identification approach which exploits the class activation value to identify the true label in candidate label sets.
- CLPL Cour et al. (2011): A avearging-based disambiguation approach based on a convex learning formulation.

Table 3: Classification accuracy (mean±std) of each comparing approach on benchmark datasets corrupted by the ID generation process.

| | MNIST | Kuzushiji-MNIST | Fashion-MNIST | CIFAR-10 | CIFAR-100 |
|---|---|---|---|---|---|
| PRODEN | 97.70±0.03% | 87.60±0.23% | 87.21±0.11% | 76.77±0.63% | 55.12±0.12% |
| PRODEN+POP | 97.87±0.04% | 88.70±0.02% | 87.62±0.04% | 79.00±0.28% | 57.68±0.14% |
| RC | 97.72±0.02% | 87.25±0.06% | 87.06±0.14% | 76.49±0.52% | 55.18±0.70% |
| RC+POP | 98.08±0.03% | 87.78±0.09% | 87.45±0.05% | 78.89±0.17% | 57.66±0.11% |
| CC | 97.25±0.11% | 83.31±0.07% | 86.01±0.13% | 72.87±0.82% | 55.56±0.23% |
| CC+POP | 97.99±0.06% | 83.98±0.10% | 86.32±0.06% | 77.03±0.58% | 56.18±0.06% |
| LW | 96.80±0.07% | 84.46±0.22% | 86.25±0.01% | 46.77±0.66% | 48.00±0.16% |
| LW+POP | 97.47±0.06% | 84.71±0.07% | 86.40±0.05% | 48.54±0.04% | 49.61±0.27% |
| CAVL | 96.25±0.40% | 79.38±0.69% | 84.66±0.05% | 62.69±1.65% | 47.35±0.16% |
| CAVL+POP | 96.71±0.11% | 79.83±0.12% | 85.04±0.10% | 63.12±0.23% | 47.61±0.06% |
| CLPL | 96.11±0.21% | 83.31±0.24% | 83.16±0.25% | 53.61±0.31% | 22.31±0.11% |
| CLPL+POP | 96.51±0.22% | 83.63±0.11% | 83.71±0.15% | 54.22±0.51% | 23.37±0.29% |

Table 4: Classification accuracy (mean±std) of each comparing approach on the real-world datasets.

| | Lost | BirdSong | MSRCv2 | Mirflickr | Malagasy | Soccer Player | Yahoo!News |
|---|---|---|---|---|---|---|---|
| PRODEN | 76.47±0.25% | 73.44±0.12% | 45.10±0.16% | 59.59±0.52% | 69.34±0.09% | 54.05±0.15% | 66.14±0.10% |
| PRODEN+POP | 78.57±0.45% | 74.47±0.36% | 45.86±0.28% | 61.09±0.10% | 72.29±0.33% | 54.48±0.10% | 66.38±0.07% |
| RC | 76.26±0.46% | 69.33±0.32% | 49.47±0.43% | 58.93±0.10% | 70.69±0.14% | 56.02±0.59% | 63.51±0.20% |
| RC+POP | 78.56±0.45% | 70.77±0.26% | 51.18±0.59% | 59.65±0.52% | 71.04±0.10% | 56.49±0.03% | 63.86±0.22% |
| CC | 63.54±0.25% | 69.90±0.58% | 41.50±0.44% | 58.81±0.54% | 69.53±0.34% | 49.07±0.36% | 54.86±0.48% |
| CC+POP | 65.47±0.93% | 71.50±0.06% | 43.21±0.43% | 59.89±0.48% | 71.19±0.40% | 49.36±0.02% | 55.22±0.05% |
| LW | 73.13±0.32% | 51.45±0.26% | 49.85±0.49% | 54.50±0.81% | 59.34±0.25% | 50.24±0.45% | 48.21±0.29% |
| LW+POP | 75.30±0.26% | 52.35±0.26% | 52.42±0.86% | 55.46±0.27% | 60.85±0.57 | 50.94±0.47% | 48.6±0.12% |
| CAVL | 73.96±0.51% | 69.63±0.93% | 46.62±1.29% | 57.13±0.10% | 65.82±0.06% | 52.92±0.40% | 60.97±0.13% |
| CAVL+POP | 75.32±0.11% | 70.13±0.22% | 46.92±0.13% | 58.63±0.48% | 67.70±0.19% | 53.44±0.10% | 61.37±0.11% |
| CLPL | 63.39±0.12% | 62.90±3.33% | 37.8±0.71% | 58.87±0.10% | 64.25±0.29% | 48.23±0.03% | 49.42±0.13% |
| CLPL+POP | 64.73±0.14% | 64.06±0.48% | 39.32±0.24% | 60.31±0.27% | 66.04±0.25% | 49.11±0.21% | 50.33±0.18% |

- PICO Wang et al. (2022b): a data-augmentation-based method which identifies the true label via contrastive-learning with learned prototypes for image datasets.
- RCR Wu et al. (2022): a data-augmentation-based method which identifies the true label via consistency regularization with random augmented instances for image datasets.

For the benchmark datasets, we use the same data augmentation strategy for the data-augmentation-free methods (VALEN, PRODEN, RC, CC, LW and CAVL) to make fair comparisons with the data-augmentation-based methods (PICO and RCR). However, data augmentation cannot be employed on the realworld datasets that contain extracted feature from audio and video data, we just compared our methods with the data-augmentation-free methods on realworld datasets.

For all the deep approaches, We used the same training/validation setting, models, and optimizer for fair comparisons. Specifically, a 5-layer LeNet is trained on MNIST, Kuzushiji-MNIST and Fashion-MNIST, the Wide-ResNet-28-2 Zagoruyko & Komodakis (2016) is trained on CIFAR-10 and CIFAR-100, and the linear model is trained on real-world PLL datasets, respectively. The hyperparameters are selected so as to maximize the accuracy on a validation set (10% of the training set). We run 5 trials on the benchmark datasets and the real-world PLL datasets. The mean accuracy as well as standard deviation are recorded for all comparing approaches. All the comparing methods are implemented with PyTorch.

## 4.3 EXPERIMENTAL RESULTS

Table 1 and Table 2 report the classification accuracy of each approach on benchmark datasets corrupted by the ID generation process and the real-world PLL datasets, respectively. Due to the inability of data augmentation to be employed on extracted feature , we didn't compare our methods with PICO and RCR on realworld datasets. The best results are highlighted in bold. We can observe

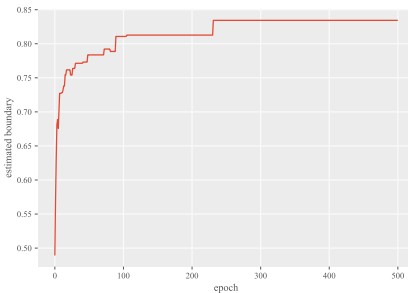

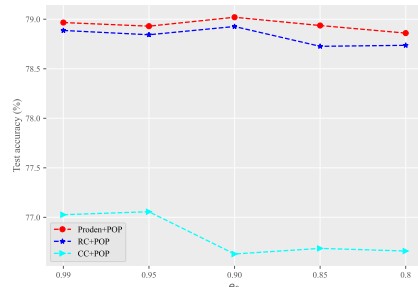

Figure 1: Estimated purified region on Lost.

Figure 2: Hyper-parameter sensitivity on CIFAR-10.

that POP achieves the best performance against other approaches in most cases and the performance advantage of POP over comparing approaches is stable under varying the number of candidate labels.

In addition, to analysis the purified region in Theorem 1, we employ the confidence predictions of $f(x, \tilde{\theta})$ (the network in Section 4.1) as the posterior and plot the curve of the estimated purified region in every epoch on Lost in Figure 1. We can see that although the estimated purified region would be not accurate enough, the curve could show that the trend of continuous increase for the purified region.

### 4.4 FURTHER ANALYSIS

As the framework of POP is flexible for the loss function, we integrate the proposed method with the previous methods for instance-independent PLL including PRODEN, RC, CC, LW, CAVL and CLPL. In this subsection, we empirically prove that the previous methods for instance-independent PLL could be promoted to achieve better performance after integrating with POP.

Table 3 and Table 4 report the classification accuracy of each method for instance-independent PLL and its variant integrated with POP on benchmark datasets corrupted by the ID generating procedure and the real-world datasets, respectively. We didn't use any data augmentation on benchmark datasets in this part of experiments. As shown in Table 3 and Table 4, the approaches integrated with POP including PRODEN+POP, RC+POP, CC+POP , LW+POP, CAVL+POP and CLPL+POP achieve superior performance against original method, which clearly validates the usefulness of POP framework for improving performance for ID PLL.

Figure 3 illustrates the variant integrated with POP performs under different hyper-parameter configurations on CIFAR-10 while similar observations are also made on other data sets. The hyperparameter sensitivity on other datasets could be founded in Appendix A.4. As shown in Figure 3, it is obvious that the performance of the variant integrated with POP is relatively stable across a broad range of each hyper-parameter. This property is quite desirable as POP framework could achieve robust classification performance.

### 5 CONCLUSION

In this paper, the problem of partial label learning is studied where a novel approach POP is proposed. we consider ID partial label learning and propose a theoretically-guaranteed approach, which could train the classifier with progressive purification of the candidate labels and is theoretically guaranteed to eventually approximates the Bayes optimal classifier for ID PLL. Experiments on benchmark and real-world datasets validate the effectiveness of the proposed method. If PLL methods become very effective, the need for exactly annotated data would be significantly reduced. As a result, the employment of data annotators might be decreased which could lead to a negative societal impact.

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

# A   APPENDIX

## A.1   PROOFS OF THEOREM 1

Assume that there exists a set $L(e)$ for all $\boldsymbol{x} \in L(e)$ which satisfies $y^{\boldsymbol{x}} = \arg\max_j f_j(\boldsymbol{x})$ and $p(y^{\boldsymbol{x}}|\boldsymbol{x}) - p(o|\boldsymbol{x}) \geq e$, we have

$$\mathbb{E}_{(\boldsymbol{z},S)\sim\tilde{p}(\boldsymbol{z},S_{\text{new}})} \left[ \mathbf{1}_{\{j\neq h^\star(\boldsymbol{z})\}} \Big| p(y^{\boldsymbol{z}}|\boldsymbol{z}) - p(o|\boldsymbol{z}) \geq p(y^{\boldsymbol{x}}|\boldsymbol{x}) - p(o|\boldsymbol{x}),\ j \in S_{\boldsymbol{z}} \right] = 0 \qquad (9)$$

Let $e_{\text{new}}$ be the new boundary and $\frac{\epsilon}{6l\alpha}(p(y^{\boldsymbol{x}}|\boldsymbol{x}) - e) \leq e - e_{\text{new}} \leq \frac{\epsilon}{3l\alpha}(p(y^{\boldsymbol{x}}|\boldsymbol{x}) - e)$. As the probability density function $d(u)$ of the margin $u(\boldsymbol{x}) = p(y^{\boldsymbol{x}}|\boldsymbol{x}) - p(o|\boldsymbol{x})$ is bounded by $c_\star <$

$d(u) < c^\star$, we have the following result for $\boldsymbol{x}$ that satisfies $p(y^{\boldsymbol{x}}|\boldsymbol{x}) - p(o|\boldsymbol{x}) \geq e_{\text{new}}$ [3]

$$
\begin{aligned}
&\mathbb{E}_{(\boldsymbol{z},S)\sim\tilde{p}(\boldsymbol{z},S_{\text{new}})} \left[\mathbf{1}_{\{j\neq h^\star(\boldsymbol{z})\}} \Big| j \in S_{\boldsymbol{z}}, p(y^{\boldsymbol{z}}|\boldsymbol{z}) - p(o|\boldsymbol{z}) \geq p(y^{\boldsymbol{x}}|\boldsymbol{x}) - p(o|\boldsymbol{x})\right] \\
\leq& \mathbb{E}_{(\boldsymbol{z},S)\sim\tilde{p}(\boldsymbol{z},S_{\text{new}})} \left[\mathbf{1}_{\{j\neq h^\star(\boldsymbol{z})\}} \Big| p(y^{\boldsymbol{z}}|\boldsymbol{z}) - p(o|\boldsymbol{z}) \geq p(y^{\boldsymbol{x}}|\boldsymbol{x}) - p(o|\boldsymbol{x})\right] \\
=& \mathbb{P}_{\boldsymbol{z}} \left[j \neq h^\star(\boldsymbol{z}) \Big| p(y^{\boldsymbol{z}}|\boldsymbol{z}) - p(o|\boldsymbol{z}) \geq p(y^{\boldsymbol{x}}|\boldsymbol{x}) - p(o|\boldsymbol{x})\right] \\
=& \frac{\mathbb{P}_{\boldsymbol{z}} \left[j \neq h^\star(\boldsymbol{z}), p(y^{\boldsymbol{z}}|\boldsymbol{z}) - p(o|\boldsymbol{z}) \geq p(y^{\boldsymbol{x}}|\boldsymbol{x}) - p(o|\boldsymbol{x})\right]}{\mathbb{P}_{\boldsymbol{z}} \left[p(y^{\boldsymbol{z}}|\boldsymbol{z}) - p(o|\boldsymbol{z}) \geq p(y^{\boldsymbol{x}}|\boldsymbol{x}) - p(o|\boldsymbol{x})\right]} \\
\leq& \frac{\mathbb{P}_{\boldsymbol{z}} \left[j \neq h^\star(\boldsymbol{z}), p(y^{\boldsymbol{z}}|\boldsymbol{z}) - p(o|\boldsymbol{z}) \geq e\right]}{\mathbb{P}_{\boldsymbol{z}} \left[p(y^{\boldsymbol{z}}|\boldsymbol{z}) - p(o|\boldsymbol{z}) \geq p(y^{\boldsymbol{x}}|\boldsymbol{x}) - p(o|\boldsymbol{x})\right]} + \frac{\mathbb{P}_{\boldsymbol{z}} \left[j \neq h^\star(\boldsymbol{z}), e_{\text{new}} \leq p(y^{\boldsymbol{z}}|\boldsymbol{z}) - p(o|\boldsymbol{z}) < e\right]}{\mathbb{P}_{\boldsymbol{z}} \left[p(y^{\boldsymbol{z}}|\boldsymbol{z}) - p(o|\boldsymbol{z}) \geq p(y^{\boldsymbol{x}}|\boldsymbol{x}) - p(o|\boldsymbol{x})\right]} \\
=& \frac{\mathbb{P}_{\boldsymbol{z}} \left[j \neq h^\star(\boldsymbol{z}), p(y^{\boldsymbol{z}}|\boldsymbol{z}) - p(o|\boldsymbol{z}) \geq e\right]}{\mathbb{P}_{\boldsymbol{z}} \left[p(y^{\boldsymbol{z}}|\boldsymbol{z}) - p(o|\boldsymbol{z}) \geq e\right]} \frac{\mathbb{P}_{\boldsymbol{z}} \left[p(y^{\boldsymbol{z}}|\boldsymbol{z}) - p(o|\boldsymbol{z}) \geq e\right]}{\mathbb{P}_{\boldsymbol{z}} \left[p(y^{\boldsymbol{z}}|\boldsymbol{z}) - p(o|\boldsymbol{z}) \geq p(y^{\boldsymbol{x}}|\boldsymbol{x}) - p(o|\boldsymbol{x})\right]} \\
& + \frac{\mathbb{P}_{\boldsymbol{z}} \left[j \neq h^\star(\boldsymbol{z}), e_{\text{new}} \leq p(y^{\boldsymbol{z}}|\boldsymbol{z}) - p(o|\boldsymbol{z}) < e\right]}{\mathbb{P}_{\boldsymbol{z}} \left[p(y^{\boldsymbol{z}}|\boldsymbol{z}) - p(o|\boldsymbol{z}) \geq p(y^{\boldsymbol{x}}|\boldsymbol{x}) - p(o|\boldsymbol{x})\right]} \\
=& \underbrace{\mathbb{E}_{(\boldsymbol{z},S)\sim\tilde{p}(\boldsymbol{z},S)} \left[\mathbf{1}_{\{h(\boldsymbol{z})\neq y^{\boldsymbol{z}}\}} \Big| p(y^{\boldsymbol{z}}|\boldsymbol{z}) - p(o|\boldsymbol{z}) \geq e\right]}_{=0(\text{According to Eq. (9)})} \frac{\mathbb{P}_{\boldsymbol{z}} \left[p(y^{\boldsymbol{z}}|\boldsymbol{z}) - p(o|\boldsymbol{z}) \geq e\right]}{\mathbb{P}_{\boldsymbol{z}} \left[p(y^{\boldsymbol{z}}|\boldsymbol{z}) - p(o|\boldsymbol{z}) \geq p(y^{\boldsymbol{x}}|\boldsymbol{x}) - p(o|\boldsymbol{x})\right]} \\
& + \frac{\mathbb{P}_{\boldsymbol{z}} \left[j \neq y^{\boldsymbol{z}}, e_{\text{new}} \leq p(y^{\boldsymbol{z}}|\boldsymbol{z}) - p(o|\boldsymbol{z}) < e\right]}{\mathbb{P}_{\boldsymbol{z}} \left[p(y^{\boldsymbol{z}}|\boldsymbol{z}) - p(o|\boldsymbol{z}) \geq p(y^{\boldsymbol{x}}|\boldsymbol{x}) - p(o|\boldsymbol{x})\right]} \\
=& \frac{\mathbb{P}_{\boldsymbol{z}} \left[e_{\text{new}} \leq p(y^{\boldsymbol{z}}|\boldsymbol{z}) - p(o|\boldsymbol{z}) < e\right]}{\mathbb{P}_{\boldsymbol{z}} \left[p(y^{\boldsymbol{z}}|\boldsymbol{z}) - p(o|\boldsymbol{z}) \geq p(y^{\boldsymbol{x}}|\boldsymbol{x}) - p(o|\boldsymbol{x})\right]} \\
\leq& \frac{c^\star(e - e_{\text{new}})}{c_\star(p(y^{\boldsymbol{x}}|\boldsymbol{x}) - e)}.
\end{aligned}
\tag{10}
$$

Due to that $\frac{\epsilon}{6l\alpha}(p(y^{\boldsymbol{x}}|\boldsymbol{x}) - e) \leq e - e_{\text{new}} \leq \frac{\epsilon}{3l\alpha}(p(y^{\boldsymbol{x}}|\boldsymbol{x}) - e)$ holds, we can further relax Eq. (10) as follows:

$$
\begin{aligned}
&\mathbb{E}_{(\boldsymbol{z},S)\sim\tilde{p}(\boldsymbol{z},S_{\text{new}})} \left[\mathbf{1}_{\{j\neq h^\star(\boldsymbol{z})\}} \Big| j \in S_{\boldsymbol{z}}, p(y^{\boldsymbol{z}}|\boldsymbol{z}) - p(o|\boldsymbol{z}) \geq p(y^{\boldsymbol{x}}|\boldsymbol{x}) - p(o|\boldsymbol{x})\right] \\
\leq& \frac{c^*(e - e_{\text{new}})}{c_*(p(y^{\boldsymbol{x}}|\boldsymbol{x}) - e)} \\
\leq& \frac{c^*}{c_*(p(y^{\boldsymbol{x}}|\boldsymbol{x}) - e)} \frac{\epsilon}{3l\alpha}(p(y^{\boldsymbol{x}}|\boldsymbol{x}) - e) \\
=& \frac{\epsilon}{3\alpha}.
\end{aligned}
\tag{11}
$$

Then, we can find that the assumption that the gap between $f_j(\boldsymbol{x})$ and $f_j^\star(\boldsymbol{x})$ should be controlled by the risk at point $\boldsymbol{z}$ implies:

$$
\begin{aligned}
&\left|f_j(\boldsymbol{x}) - f_j^\star(\boldsymbol{z})\right| \\
\leq& \alpha\mathbb{E}_{(\boldsymbol{z},S)\sim\tilde{p}(\boldsymbol{z},S_{\text{new}})} \left[\mathbb{1}_{\{h(\boldsymbol{z})\neq y^{\boldsymbol{z}}\}} \Big| p(y^{\boldsymbol{z}}|\boldsymbol{z}) - p(o|\boldsymbol{z}) \geq p(y^{\boldsymbol{x}}|\boldsymbol{x}) - p(o|\boldsymbol{x})\right] + \frac{\epsilon}{6} \\
\leq& \alpha\frac{\epsilon}{3\alpha} + \frac{\epsilon}{6} \\
\leq& \frac{\epsilon}{2}.
\end{aligned}
\tag{12}
$$

Hence, for $\boldsymbol{x}$ s.t. $p(y^{\boldsymbol{x}}|\boldsymbol{x}) - p(o|\boldsymbol{x}) \geq e_{\text{new}}$, according to Eq. (12) we have

$$
\begin{aligned}
f_{y^{\boldsymbol{x}}}(\boldsymbol{x}) - f_{j\neq y^{\boldsymbol{x}}}(\boldsymbol{x}) \geq& (p(y = y^{\boldsymbol{x}}|\boldsymbol{x}) - \frac{\epsilon}{2}) - (p(y = j|\boldsymbol{x}) + \frac{\epsilon}{2}) \\
=& p(y = y^{\boldsymbol{x}}|\boldsymbol{x}) - p(y = j|\boldsymbol{x}) - \epsilon \\
\geq& p(y = y^{\boldsymbol{x}}|\boldsymbol{x}) - p(o|\boldsymbol{x}) - \epsilon \\
\geq& e_{\text{new}} - \epsilon \\
\geq& 0,
\end{aligned}
\tag{13}
$$

---

[3]Details of Eq. (3) in the paper submission

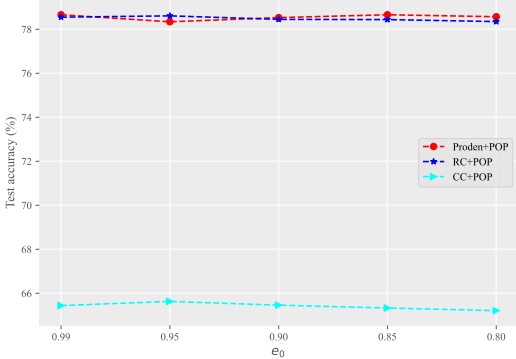

Figure 3: Hyper-parameter sensitivity on Lost.

which means that $_j(\boldsymbol{x})$ will be the same label as $h^\star$ and thus the level set $L(e_{\text{new}})$ is pure for $f$. Meanwhile, the choice of $e_{\text{new}}$ ensures that

$$
\begin{aligned}
p(y^{\boldsymbol{x}}|\boldsymbol{x}) - e_{\text{new}} &\geq p(y^{\boldsymbol{x}}|\boldsymbol{x}) - (e - \frac{\epsilon}{6l\alpha}(p(y^{\boldsymbol{x}}|\boldsymbol{x}) - e)) \\
&= p(y^{\boldsymbol{x}}|\boldsymbol{x}) - e + \frac{\epsilon}{6l\alpha}(p(y^{\boldsymbol{x}}|\boldsymbol{x}) - e) \\
&= (1 + \frac{\epsilon}{6l\alpha})(p(y^{\boldsymbol{x}}|\boldsymbol{x}) - e).
\end{aligned}
\tag{14}
$$

Here, the proof of Theorem 1 has been completed.

## A.2 DETAILS OF EQ. (5)

If $f_m(\boldsymbol{x}) - f_{j \neq m} \geq e + \epsilon$, according to Eq. (12) we have:

$$
\begin{aligned}
p(y^{\boldsymbol{x}}|\boldsymbol{x}) &\geq p(y = m|\boldsymbol{x}) \\
&= p(y = j|\boldsymbol{x}) + p(y = m|\boldsymbol{x}) - p(y = j|\boldsymbol{x}) \\
&\geq p(y = j|\boldsymbol{x}) + p(y = m|\boldsymbol{x}) - p(y = j|\boldsymbol{x}) \\
&\geq p(y = j|\boldsymbol{x}) + (f_m(\boldsymbol{x}) - \frac{\epsilon}{2}) - (f_j(\boldsymbol{x}) + \frac{\epsilon}{2}) \\
&= p(y = j|\boldsymbol{x}) + (f_m(\boldsymbol{x}) - f_j(\boldsymbol{x})) - \epsilon \\
&\geq p(y = j|\boldsymbol{x}) + (e + \epsilon) - \epsilon \\
&= p(y = j|\boldsymbol{x}) + e.
\end{aligned}
\tag{15}
$$

## A.3 PROOFS OF THEOREM 2

To begin with, we prove that there exists at least a level set $L(e_0)$ pure to $f_0$. Considering $\boldsymbol{x}$ satisfies $p(y^{\boldsymbol{x}}|\boldsymbol{x}) - p(o|\boldsymbol{x}) \geq e_0$, we have $\mathbb{P}_{\boldsymbol{z}}\left[j \neq h^\star(\boldsymbol{z}) \middle| j \in S_{\boldsymbol{z}}, p(y^{\boldsymbol{z}}|\boldsymbol{z}) - p(o|\boldsymbol{z}) \geq e_0\right] \leq p(y^{\boldsymbol{z}}|\boldsymbol{z}) - e_0 + \xi^j(\boldsymbol{z})$. Due to the assumption $|f_j(\boldsymbol{x}) - f_j^\star(\boldsymbol{x})| \leq \alpha\mathbb{E}_{(\boldsymbol{z},S)\sim\tilde{p}(\boldsymbol{z},S)}\left[\mathbf{1}_{\{j\neq h^\star(\boldsymbol{z})\}} \middle| j \in S_{\boldsymbol{z}}, p(y^{\boldsymbol{z}}|\boldsymbol{z}) - p(o|\boldsymbol{z}) \geq p(y^{\boldsymbol{x}}|\boldsymbol{x}) - p(o|\boldsymbol{x})\right] + \frac{\epsilon}{6}$, it suffices to satisfy $\alpha(p(y^{\boldsymbol{x}}|\boldsymbol{x}) - e_0 + \xi) + \frac{\epsilon}{6} \leq e_0$ to ensure that $f_j(\boldsymbol{x})$ has the same prediction with $h^\star$ when $p(y^{\boldsymbol{x}}|\boldsymbol{x}) - p(o|\boldsymbol{x}) \geq e_0$. Since we have $\xi^j(\boldsymbol{x}) \leq p(y = j|\boldsymbol{x})t \leq p(y^{\boldsymbol{x}}|\boldsymbol{x})t$, by choosing $e_0 \geq \frac{(1+t)\alpha + \frac{\epsilon}{6}}{1+\alpha} \geq \frac{(1+t)\alpha p(y^{\boldsymbol{x}}|\boldsymbol{x}) + \frac{\epsilon}{6}}{1+\alpha}$ one can ensure that initial $f_0$ has a pure $L(e_0)$-level set.

Then in the rest of the iterations we ensure the level set $p(y^{\boldsymbol{z}}|\boldsymbol{z}) - p(o|\boldsymbol{z}) \geq e$ is pure. We decrease $e$ by a reasonable factor to avoid incurring too many corrupted labels while ensuring enough progress in label purification, i.e. $\frac{\epsilon}{6l\alpha}(p(y^{\boldsymbol{x}}|\boldsymbol{x}) - e) \leq e - e_{\text{new}} \leq \frac{\epsilon}{3l\alpha}(p(y^{\boldsymbol{x}}|\boldsymbol{x}) - e)$, such that in the

Table 5: Characteristic of the benchmark datasets corrupted by the ID generation process.

| Dataset | #Train | #Test | #Features | #Class Labels | avg. #CLs |
|---|---|---|---|---|---|
| MNIST | 60000 | 10000 | 784 | 10 | 8.71 |
| Fashion-MNIST | 60,000 | 10,000 | 784 | 10 | 3.46 |
| Kuzushiji-MNIST | 60,000 | 10,000 | 784 | 10 | 3.87 |
| CIFAR-10 | 50,000 | 10,000 | 3,072 | 10 | 3.68 |
| CIFAR-100 | 50,000 | 10,000 | 3,072 | 100 | 4.64 |

Table 6: Characteristic of the real-world PLL datasets.

| Dataset | #Train | #Test | #Features | #Class Labels | avg. #CLs | Task Domain |
|---|---|---|---|---|---|---|
| Lost | 898 | 224 | 108 | 16 | 2.23 | automatic face naming Cour et al. (2011) |
| MSRCv2 | 1,406 | 352 | 48 | 23 | 3.16 | object classification Liu & Dietterich (2012) |
| Mirflickr | 2224 | 556 | 1536 | 14 | 2.76 | web image classification Huiskes & Lew (2008) |
| BirdSong | 3,998 | 1,000 | 38 | 13 | 2.18 | bird song classification Briggs et al. (2012) |
| Malagasy | 4243 | 1069 | 384 | 44 | 8.35 | POS Tagging Garrette & Baldridge (2013) |
| Soccer Player | 13,978 | 3,494 | 279 | 171 | 2.09 | automatic face naming Zeng et al. (2013) |
| Yahoo! News | 18,393 | 4,598 | 163 | 219 | 1.91 | automatic face naming Guillaumin et al. (2010) |

level set $p(y^{\boldsymbol{x}}|\boldsymbol{x}) - p(o|\boldsymbol{x}) \geq e_{\text{new}}$ we have $|f_j(\boldsymbol{x}) - f_j^{\star}(\boldsymbol{x})| \leq \frac{\epsilon}{2}$. This condition ensures the correctness of flipping when $e \geq \epsilon$. The the purified region cannot be improved once $e < \epsilon$ since there is no guarantee that $f_j(\boldsymbol{x})$ has consistent label with $h^{\star}$ when $p(y^{\boldsymbol{x}}|\boldsymbol{x}) - p(o|\boldsymbol{x}) < \epsilon$ and $|f_j(\boldsymbol{x}) - f_j^{\star}(\boldsymbol{x})| \leq \frac{\epsilon}{2}$. To get the largest purified region, we can set $e_{\text{end}} = \epsilon$. Since the probability density function $d(u)$ of the margin $u(\boldsymbol{x}) = p(y^{\boldsymbol{x}}|\boldsymbol{x}) - p(o|\boldsymbol{x})$ is bounded by $c_{\star} \leq d(u) \leq c^{\star}$, we have:

$$
\begin{aligned}
\mathbb{P}_{\boldsymbol{x} \sim D}[y_{f_{final}(\boldsymbol{x})} \neq h^{\star}] &\leq \mathbb{P}[p(y^{\boldsymbol{x}}|\boldsymbol{x}) - p(o|\boldsymbol{x}) < e_{\text{end}}] \\
&= \mathbb{P}_{\boldsymbol{x} \sim D}[p(y^{\boldsymbol{x}}|\boldsymbol{x}) - p(o|\boldsymbol{x}) < \epsilon] \\
&\leq c^{\star}\epsilon.
\end{aligned}
\tag{16}
$$

Then $\mathbb{P}_{\boldsymbol{x} \sim D}[y_{f_{final}(\boldsymbol{x})} = h^{\star}] = 1 - \mathbb{P}_{\boldsymbol{x} \sim D}[y_{f_{final}(\boldsymbol{x})} \neq h^{\star}] \geq 1 - c^{\star}\epsilon.$

The rest of the proof is the total round $R \geq \frac{6\alpha l}{\epsilon} \log(\frac{1-\epsilon}{\frac{1}{c}-e_0})$, which follows from the fact that each round of label flipping improves the the purified region by a factor of $(1 + \frac{\epsilon}{6l\alpha})$:

$$
\begin{aligned}
&\left(1 + \frac{\epsilon}{6l\alpha}\right)^R (p(y^{\boldsymbol{x}}|\boldsymbol{x}) - e_0) \geq p(y^{\boldsymbol{x}}|\boldsymbol{x}) - \epsilon \\
\Rightarrow &\left(1 + \frac{\epsilon}{6l\alpha}\right)^R \geq \frac{p(y^{\boldsymbol{x}}|\boldsymbol{x}) - \epsilon}{p(y^{\boldsymbol{x}}|\boldsymbol{x}) - e_0} \\
\Rightarrow &R \log\left(1 + \frac{\epsilon}{6l\alpha}\right) \geq \log\left(\frac{p(y^{\boldsymbol{x}}|\boldsymbol{x}) - \epsilon}{p(y^{\boldsymbol{x}}|\boldsymbol{x}) - e_0}\right) \\
\Rightarrow &R\frac{\epsilon}{6l\alpha} \geq R \log\left(1 + \frac{\epsilon}{6l\alpha}\right) \geq \log\left(\frac{p(y^{\boldsymbol{x}}|\boldsymbol{x}) - \epsilon}{p(y^{\boldsymbol{x}}|\boldsymbol{x}) - e_0}\right) \\
\Rightarrow &R \geq \frac{6l\alpha}{\epsilon} \log\left(\frac{p(y^{\boldsymbol{x}}|\boldsymbol{x}) - \epsilon}{p(y^{\boldsymbol{x}}|\boldsymbol{x}) - e_0}\right) \geq \frac{6l\alpha}{\epsilon} \log(\frac{1-\epsilon}{\frac{1}{c}-e_0}).
\end{aligned}
\tag{17}
$$

## A.4 DETAILS OF EXPERIMENTS

We collect four widely used benchmark datasets including MNIST LeCun et al. (1998), Kuzushiji-MNIST Clanuwat et al. (2018), Fashion-MNIST Xiao et al. (2017), CIFAR-10 Krizhevsky & Hinton (2009), CIFAR-100 Krizhevsky & Hinton (2009). In addition, five real-world PLL datasets are adopted, which are collected from several application domains including `Lost` Cour et al. (2011), `Soccer Player` Zeng et al. (2013) and `Yahoo!News` Guillaumin et al. (2010) for automatic face naming from images or videos, `MSRCv2` Liu & Dietterich (2012) for object classification, and

`BirdSong` Briggs et al. (2012) for bird song classification. Figure 3 illustrates the variant integrated with POP performs under different hyper-parameter configurations on Lost.

The average number of candidate labels (avg. #CLs) for each benchmark dataset corrupted by the ID generation process is recorded in Table-5 and the average number of candidate labels (avg. #CLs) for each real-world PLL dataset is recorded in Table-6.

