# OpenReview forum: "Progressive Purification for Instance-Dependent Partial Label Learning"
_ICLR.cc/2023/Conference — Submitted to ICLR 2023_

### Official Review · Reviewer_K87B · 2022-10-20

**Confidence:** 4
**Clarity, Quality, Novelty And Reproducibility:** Hard to evaluate quality because clar…
**Correctness:** 2
**Technical Novelty And Significance:** 2
**Empirical Novelty And Significance:** 2
**Recommendation:** 3

**Strength And Weaknesses:**

This paper presumably has multiple good ideas embedded in it, but the exposition is so poor that this reviewer just gave up trying to understand it.

The trouble really starts in section 3.3:
  * the indicator function is introduced as equivalent to "inconsistent with the Bayes classifier" but it is unclear what this means.
  * then equation (1) is presented, which is highly confusing.
    * On the left hand side is "f^j" (also sometimes called "f_j" in the text ...), but what is "f"?  The reader can eventually deduce, if they make it as far as equation (6), that "f" means "the prediction of some model in the model class which is the resulting of training on the data".
    * On the right hand side are quantities that are problem dependent but not current model dependent.  (I think?  It's hard to tell ...)
    * So, without remark, equation (1) appears to require a strong uniform property over the hypothesis class.
  * then we have some unclear terminology on "density of the margin" leading to a bound on a heretofore undefined quantity "density-imbalance ratio" (what is that?)
  * finally we arrive at Definition 1, which is the first clear thing said in section 3.3.
  * **constructive feedback**: rewrite all of section 3.3 to be as clear as definition 1, don't try to save space, and get some help from somebody with good english language writing skills to proofread.

Then there's Theorem 1.  It talks about an updated data distribution, but the associated Algorithm 1 reuses the same data set.  Only the first pass over this data set can be considered an IID sample from the true distribution.  Every subsequent pass is a data-dependent quantity and no longer IID.  **constructive feedback**: 1) rewrite algorithm 1 assuming access to more data and then consume the data as a stream and filter it using the current margin condition so that it really is an IID draw from a distribution and 2) modify theorem 1 to align with the modified algorithm 1 and then 3) if you don't want to redo all your experiments just say "we depart from the theory by reusing the same fixed dataset over and over, but the empirics are reasonable".  (The truth is, experiments often depart from the theory in some way, as long as you are clear on this it doesn't bother this reviewer).

At this point the reviewer just stopped (actually truthfully I peeked at Tables 1 and 2 and the lifts seemed rather modest and then stopped).  The meta-feedback is, if you don't have clarity, you will lose the modern reviewer, because the quantity of papers to review has gone up and the quality has gone down, so patience is at a historic minimum.

**Summary Of The Paper:**

Authors propose and test an algorithm for the instance-dependent partial label learning setting.

**Summary Of The Review:**

It needs a significant rewrite and Theorem 1 coupled with Algorithm 1 is, as stated, incorrect.

---

### Official Review · Reviewer_T9T6 · 2022-10-23

**Confidence:** 4
**Correctness:** 4
**Technical Novelty And Significance:** 3
**Empirical Novelty And Significance:** 3
**Recommendation:** 3

**Clarity, Quality, Novelty And Reproducibility:**

This paper is well-organized and well-written. The proposed method is easy to follow, and the Reproducibility could be ensured.

**Strength And Weaknesses:**

Strengths:
1. This paper provides a new perspective on improving the accuracy of instance-dependent partial label learning.
2. The proposed approach can be guaranteed to enlarge the region and approximates the Bayes optimal classifier with mild assumptions, which is the first theoretically guaranteed approach for instance-dependent partial label learning.
3. The proposed approach is flexible with arbitrary losses and compatible with deep networks, so that the previous advanced losses for partial label learning can be embedded in it and the performance is often significantly improved.
4. The theoretical justifications and empirical validations are strong. I believe this work is solid.

Weaknesses:
1. This paper could give more discussion about the reason why adopt the PLL loss in Eq. (7) to initialize the model.
2. It would be nice to validate the proposed method on more real-world partial label learning datasets.
3. The authors should give more details about how to estimate the purified region in every epoch in Figure 1.


**Summary Of The Paper:**

This paper proposes a theoretically grounded and practically effective approach to deal with the instance-dependent partial label learning problem. This paper updates the learning model while purifying each PL for the next epoch of the model training by progressively moving out false candidate labels. Theoretically, the authors prove that the proposed approach enlarges the region appropriately fast where the model is reliable, and eventually approximates the Bayes optimal classifier with mild assumptions. In addition, the proposed approach is flexible with arbitrary losses and compatible with deep networks, so that the previous advanced losses for partial label learning can be embedded in it and the performance is often significantly improved. Experimental results on various datasets validate the effectiveness of the proposed method.

**Summary Of The Review:**

This paper is novel and solid, and the theoretical guarantee is a nice result.

---

### Official Review · Reviewer_T1k7 · 2022-10-25

**Confidence:** 3
**Correctness:** 3
**Technical Novelty And Significance:** 3
**Empirical Novelty And Significance:** 2
**Recommendation:** 5

**Clarity, Quality, Novelty And Reproducibility:**

Overall, the writing quality is good. However, I think that the assumption used for theoretical analysis might be strong. The classifier trained with instance-dependent partial label learning can approximate the Bayes optimal classifier sounds an ambitious task. To yield the identifiability of the noise, intuitively, some strong assumptions have to be made.

**Details Of Ethics Concerns:**

I have not found any ethics concerns.

**Strength And Weaknesses:**

Strength:
+ The research problem is important and may have many practical applications. The generative of partial labels in real-world datasets may depend on instances. How to improve the robustness of learning models in this setting can be a challenging and important research problem.
+ The authors have shown that the proposed estimator can approximate the Bayes optimal classifier under some assumptions.
+ This paper generally is well-written and easy to follow.
Weakness:
+ The theoretical guarantee rely on the assumption that there exists a boundary $e$ for all $x$ which satisfies $y^x$ = $\arg\max_j f_j(x)$ and $p(y^x|x)− p(o|x) \geq e$. It seems that this assumption can be a little bit strong. I think it would be great to add some justification.
+ The empirical improvement of the proposed method on real-world datasets seems not large.
+ It seems that this paper is related to the paper: “Progressive Identification of True Labels for Partial-Label Learning”. It would be great to provide some justifications from the perspective of the major technical contribution.


**Summary Of The Paper:**

In this paper, the authors have proposed a method for instance-dependent partial-label learning with theoretical guarantees. The method progressively moves out false candidate labels during the training. The experimental results demonstrate the effectiveness of the proposed method.

**Summary Of The Review:**

My largest concern is the assumption used for theoretical analysis. It would be great if the authors can make some justification for the assumption that that when will exist a boundary $e$ for all $x$ which satisfies $y^x$ = $\arg\max_j f_j(x)$ and $p(y^x|x)− p(o|x) \geq e$.

---

### Official Review · Reviewer_s5av · 2022-10-25

**Confidence:** 3
**Correctness:** 4
**Technical Novelty And Significance:** 4
**Empirical Novelty And Significance:** 4
**Recommendation:** 5

**Clarity, Quality, Novelty And Reproducibility:**

The paper is clearly written, and the experimental and theoretical results are well organized. Removing incorrect labels from the partial label set with statistical guarantees is also new in the field of partial-label learning. This method can be easily reproduced by traversing the training set in each epoch after updating the model.

**Strength And Weaknesses:**

Strengths:
1. The implementation of the proposed method is described in detail and all the claims in this paper are provided with theoretical justification. Experimental results also show that it can enhance the performance of existing PLL methods.
2. The proposed Pop method focuses on updating the partial-label set, which does not rely on the update of the model. This indicates that it can be combined with any partial-label learning method.

Weaknesses:
1. In Section 3.1, it is indicated that a score-based classifier $h(x)$ that achieves the best accuracy can recover the class-posterior probability after a softmax transformation. It is not true if there are no restrictions on the type of used surrogate loss.

2. In Section 2, the related works are split into two different categories: deep and non-deep PLL. However, I think the method CLPL [1] can also be combined with deep models since it makes no assumption on the type of model. I think the authors should discuss more CLPL and add the experimental results of it as in the previous work of deep PLL [2].

[1]. T. Cour, B. Sapp, and B. Taskar. Learning from partial labels. Journal of Machine Learning Research, 12(5):1501–1536, 2011.

[2]. L. Feng, J. Lv, B. Han, M. Xu, G. Niu, X. Geng, B. An, and M. Sugiyama. Provably consistent partial-label learning. In Advances in Neural Information Processing Systems 33 (NeurIPS’20),pp. 10948–10960, 2020b.

**Summary Of The Paper:**

This paper is proposed to achieve better performance both practically and theoretically on the problem of Instance-Dependent Partial Label Learning. To be more detailed, a purification strategy is proposed in this paper. This method works in an alternative update manner on both model and dataset: in each round, the model is first updated using the partially labeled dataset, and then a purification strategy is applied to remove labels that are regarded as 'incorrect' in each data point' partial label set. With mild assumptions, the authors show that their method is statistically consistent. Experimental results also show the efficiency of their method.

**Summary Of The Review:**

This paper proposed a method for instance-dependent partial-label learning that is valid both theoretically and practically.

---

### Decision · Program_Chairs · 2023-01-20

**Decision:**

Reject

**Justification For Why Not Higher Score:**

The paper claims a theoretical result as a main contribution. There appears however to be an error in the formulation of this result.

**Justification For Why Not Lower Score:**

N/A

**Metareview: Summary, Strengths And Weaknesses:**

The paper proposes a technique for partial label learning, wherein samples have an associated subset of labels of varying sizes. The basic idea is to perform "progressive purification" via an iterative scheme, which tries to winnow down the set of possible labels.

While the idea is interesting, the paper also claims theirs is the first theoretically grounded method for the problem. Unfortunately, one of the reviewers noted that the theoretical analysis appears flawed in its current form: since the method is iterative, it requires multiple passes over the data, using previously iterates' models to change the samples. This would appear to break the IID assumption. The author response did not resolve this issue. From my reading, I concur with the reviewer's concern.

Following the response, other reviews also agreed with the concern, and lowered their scores. Thus, we do not believe the paper is currently ready for publication.

**Summary Of Ac-Reviewer Meeting:**

N/A